# Association between dietary intake of flavonoid and chronic kidney disease in US adults: Evidence from NHANES 2007-2008, 2009-2010, and 2017-2018

Peijia Liu[ID][⊙], Wujian Peng[⊙], Feng Hu, Guixia Li[ID]*

Department of Nephrology, Shenzhen Third People's Hospital, The Second Affiliated Hospital of Southern University of Science and Technology, Shenzhen, Guangdong, China

⊙ These authors contributed equally to this work.

* liguixia120@163.com

## Abstract

### Background

Studies investigating the relationship between flavonoid intake and chronic kidney disease (CKD) are limited. This study investigated the association between daily flavonoid intake and CKD in US adults by using data for 2007–2008, 2009–2010, and 2017–2018 from the National Health and Nutrition Examination Survey (NHANES) database.

### Methods

This study employed a cross-sectional design and used data from three cycles of the continuous NHANES: 2007–2008, 2009–2010, and 2017–2018. NHANES researchers collected data related to consumption of various food and beverages from participants by employing 24-h dietary recall questionnaires. CKD is defined as an estimated glomerular filtration rate of < 60 mL/min/1.73m$^2$ or a urine albumin-to-creatinine ratio of $\geq$ 30 mg/g.

### Results

The odds ratios (OR) for CKD risk in the second (Q2), third (Q3), and fourth (Q4) quartiles of total flavonoid intake, compared with that in the first (Q1) quartile, were 0.780 (95% CI: 0.600, 1.015), 0.741 (95% CI: 0.573, 0.957), and 0.716 (95% CI: 0.554, 0.925), respectively (with a $P$ value for the trend of 0.040). According to the restricted cubic spline analysis, total flavonoid intake exhibited a non-linear relationship with CKD risk (P < 0.001).

### Conclusion

Our findings suggest that a potential J-shaped relationship was observed between total flavonoid consumption and CKD risk, with an inflection point at 69.58 mg/d. Our study indicates that a moderate intake of flavonoids may confer renal benefits which may offer novel strategies for CKD treatment.

**Data Availability Statement:** The data relevant to this study are from the National Health and Nutrition Examination Survey (NHANES). NHANES data is publicly available and can be accessed at

https://www.cdc.gov/nchs/nhanes. The authors confirm they did not have any special access privileges that others would not have and others would be able to access these data in the same manner as the authors.

**Funding:** This work was supported by the Project funded by Shenzhen Third People's Hospital (No. G2022030) and the Shenzhen Science and Technology Planning Project (No. JCYJ20220530163003007). Guixia Li (corresponding author) is responsible for the project funded by Shenzhen Third People's Hospital (No. G2022030), and Wujian Peng (co-first author) is responsible for the Shenzhen Science and Technology Planning Project (No. JCYJ20220530163003007). The contributions of the authors are as follows: • Wujian Peng: Data curation, Formal analysis, Investigation, Methodology, Software, Writing - Review & Editing. • Guixia Li: Conceptualization, Funding acquisition, Project administration, Resources, Supervision, Writing - Review & Editing.

**Competing interests:** The authors have declared that no competing interests exist.

## Introduction

Chronic kidney disease (CKD) is a global public health challenge, prompting urgent widespread attention worldwide. According to data from the Chronic Kidney Disease Epidemiology Collaboration (CKD-EPI), approximately 700 million individuals globally are afflicted by CKD, and this number continues to increase steadily [1]. The CKD burden encompasses a spectrum of health and economic problems, including cardiovascular complications, renal failure, increased healthcare expenditures, diminished quality of life, socioeconomic disparities, and societal repercussions [2, 3]. Consequently, offering timely and efficacious intervention for CKD is extremely crucial. During CKD management, interventions targeting risk factors, including blood pressure, blood glucose, and uric acid levels, and dietary interventions, such as restricting protein, purine, salt, and potassium intake, are both indispensable [4]. Although these interventions offer some renal benefits to CKD patients, effective treatments are lacking.

Flavonoids are naturally occurring polyphenolic compounds found in plants. Structurally, they include aromatic rings, hydrophilic groups, and unsaturated bonds, and are subcategorized into flavones, flavonols, flavanones, isoflavones, flavan-3-ols, and anthocyanins [5–8]. Some studies have reported that high flavonoid consumption is negatively associated with cardiovascular diseases, metabolic syndrome, non-alcoholic fatty liver disease, liver fibrosis, high uric acid levels, tumors, and cognitive impairment [9–15]. This may be attributable to the biological activities of flavonoid-rich foods, such as reduction in oxidative damage, attenuation of inflammatory responses, antifibrotic effects, immunomodulation, and enhancement of cytokine secretion [5, 16–19]. Some clinical studies have also indicated the potential benefits of flavonoid-rich foods in CKD patients. A correlation study demonstrated that a high intake of flavonoid-rich foods is associated with a reduced incidence of diabetic nephropathy [20]. In a randomized double-blind trial conducted among end-stage kidney disease patients, excessive daily intake of flavanols, a type of flavonoid, improved vascular endothelial function [21]. Furthermore, renal benefits associated with flavonoid-rich foods may be a result of the collective interaction of multiple mechanisms. These signaling pathways can ameliorate glomerular hyperfiltration, decrease proteinuria, exert antioxidant effects, reduce the expression of inflammatory factors, and safeguard endothelial cells, ultimately contributing to the improvement in renal function [6, 22–26]. Despite the potential renal benefits conferred on CKD patients through the consumption of flavonoid-rich foods, large-scale population studies investigating the relationship between flavonoid consumption and CKD are lacking. Consequently, a significant research remains imperative in investigating the interplay of flavonoid intake and CKD within a broader population context.

The National Health and Nutrition Examination Survey (NHANES) database is operated by the National Center for Health Statistics (NCHS) in the US. It is a national-level health and nutrition survey project that assesses health, nutrition, and prevalence of chronic diseases among US population. We here explored the association between flavonoid consumption and CKD by using cross-sectional data from the NHANES database. Because the biological activities of different flavonoid subtypes potentially vary, we also investigated the relationship between CKD and flavonoid subgroups.

## Materials and methods

### Study design and study population

The NHANES database is a publicly accessible repository of data of study participants recruited biennially by using a complex multi-stage stratified sampling design. The repository includes demographic, questionnaire, dietary, physical examination, and laboratory

examination data. After these data are subjected to weighting procedures, they can effectively represent the entire US population [27].

Weighting is a crucial step in the analysis of NHANES data. It involves adjusting the data to account for the survey's complex sampling design, including oversampling of certain populations, survey non-responses, and post-stratification to match the US population demographics. This process ensures that the results can be generalized to the entire US population. The weighting factor provided by NHANES is applied to each participant to make the sample representative of the broader population.

This study employed a cross-sectional design and used data from three cycles of the continuous NHANES: 2007–2008, 2009–2010, and 2017–2018. After applying these weights, the results of this study are representative of 306,640,845 US residents. Following the exclusion of individuals under the age of 20 and those with missing data, a total of 23,259,068 participants were selected.

Each participant in NHANES provided written informed consent, and survey data collection was approved by the Research Ethics Review Board of the NCHS. The data used for these analyses were fully de-identified and made publicly available, and thus were exempted from the review of our institutional ethics review board.

## Dietary flavonoid intake

The total flavonoid intake from food was calculated using a complex methodology. In 2007–2010 and 2017–2018, NHANES researchers collected data related to consumption of various food and beverages from participants by employing 24-h dietary recall questionnaires. These data covered flavonoid-rich foods and beverages rich, such as onions, celery, tea, and red wine. Subsequently, food codes of Food and Nutrient Database for Dietary Studies (FNDDS) were used to match flavonoid-containing foods and acquire their respective flavonoid content. Specifically, FNDDS version 4.1 food codes were used for the NHANES 2007–2008 data, while FNDDS version 5.0 food codes were used for the NHANES 2009–2010 and 2017–2018 data [14, 28]. Daily flavonoid intake is calculated as the sum of the flavonoid content from various foods. We computed the daily flavonoid intake as the average of flavonoid intake over 2 days to provide a more specific reflection of individuals' dietary habits. Of note, the Automated Multiple-Pass Method (AMPM) utilized in the questionnaire survey has undergone comprehensive validation in numerous research studies, establishing it as a dependable approach for evaluating both group energy intake and adult sodium intake [29, 30]. Furthermore, the NHANES database provides the daily intake data of six flavonoid subgroups, namely anthocyanins, flavan-3-ols, flavanones, flavones, flavonols, and isoflavones, to facilitate a comprehensive understanding of flavonoid compounds [12, 14, 20, 31].

## Study outcomes

A solid-phase fluorescent immunoassay was conducted for measuring human urinary albumin levels [32]. The Jaffe rate method was employed to determine serum and urine creatinine concentrations. The estimated glomerular filtration rate (eGFR) was calculated using the 2012 CKD-EPI equation, which is based on serum creatinine levels [33]. According to the Kidney Disease Outcomes Quality Initiative (KDOQI) guidelines, CKD was defined as an eGFR < 60 ml/min/1.73 m$^2$ and/or a urinary albumin-to-creatinine ratio (uACR) $\geq$ 30 mg/g [4]. An eGFR < 60 ml/min/1.73 m$^2$ indicated a decline in eGFR, while a uACR $\geq$ 30 mg/g indicated proteinuria. Our primary objective was to investigate the relationship between dietary flavonoid intake and the risk of CKD in U.S. adults. The secondary endpoints were the relationships between dietary flavonoid intake and the risks of eGFR decline and proteinuria, respectively.

## Covariates

The following covariates were incorporated in this study: sex (male vs. female), age, race (Black, White, Mexican, others), education level (high school or above vs. middle school or below), height, weight, body mass index (BMI), diastolic blood pressure (DBP), systolic blood pressure (SBP), poverty income ratio (PIR), smoking status (former, never, or now), and presence of chronic diseases including hypertension, hyperlipidemia, hyperuricemia, and diabetes. Energy intake refers to the daily calorie intake (kcal). Physical activity was evaluated in terms of metabolic equivalent of task (MET) minutes per week and categorized into two groups based on whether it exceeded 600 MET minutes per week [34]. BMI is calculated by dividing weight (in kilograms) by the square of height (in meters). PIR is transformed into a binary variable with a cut-off point set at 1. Blood pressure data were based on the average of three measurements. Hypertension was defined as SBP > 140 mmHg, DBP > 90 mmHg, use of antihypertensive medications, or a physician made clinical diagnosis of hypertension [35]. Dyslipidemia is defined as hypertriglyceridemia $\geq$ 150 mg/dL, total cholesterol $\geq$ 200 mg/dL, low-density lipoprotein cholesterol (LDL) $\geq$ 130 mg/dL, high-density lipoprotein (HDL) cholesterol < 40 mg/dL in men, HDL < 50 mg/dL in females, use of lipid-lowering medications, or a physician made diagnosis of hyperlipidemia [36]. Diabetes is defined as glycated hemoglobin > 6.5%, fasting blood glucose > 7.0 mmol/L, random blood glucose > 11.1 mmol/L, 2-h oral glucose tolerance test blood glucose > 11.1mmol/L, use of antidiabetics, or a physician made clinical diagnosis of diabetes [37]. Hyperuricemia is defined as serum uric acid levels >420 μmol/L in men or >360 μmol/L in women [38].

## Statistical methods

NHANES data were obtained through a multistage stratified sampling process. Therefore, all data were weighted before data analysis to represent the entire US population. Continuous variables were presented using means and 95% confidence intervals (95% CIs), and statistical differences between the CKD and non-CKD groups for these variables were assessed using the rank-sum test. On the other hand, categorical variables were expressed as percentages with 95% CI and were compared using the chi-square test. Flavonoid intake was categorized into four quartiles: Q1, Q2, Q3, and Q4. We here explored the relationship between CKD and flavonoid intake through logistic regression. Based on the differences between the CKD and non-CKD groups and clinical experience, age, sex, BMI, ethnicity, education level, drinking status, smoking status, hyperuricemia, hyperlipidemia, hypertension, and diabetes mellitus were selected as covariates in the weighted logistic model. A restricted cubic spline (RCS) was employed for assessing the non-linear relationship between flavonoid intake and CKD risk. Furthermore, relationship between dietary flavonoid consumption and CKD were investigated across subgroup analyses, with interaction tests. Finally, logistic regression models were used to examine the relationship between CKD and different flavonoid subclasses. A $P < 0.05$ (two-sided) significance level was used to indicate statistical significance. All statistical analyses were conducted using R version 4.3.0 (R Development Core Team, University of Auckland, Auckland City, NZ).

## Results

### Characteristics of the population

The NHANES data is weighted to account for the complex survey design, non-response, and post-stratification adjustments. This ensures that the findings are representative of the total US population. Post-weighting, the data in this study represented 213,259,068 US adults aged 20

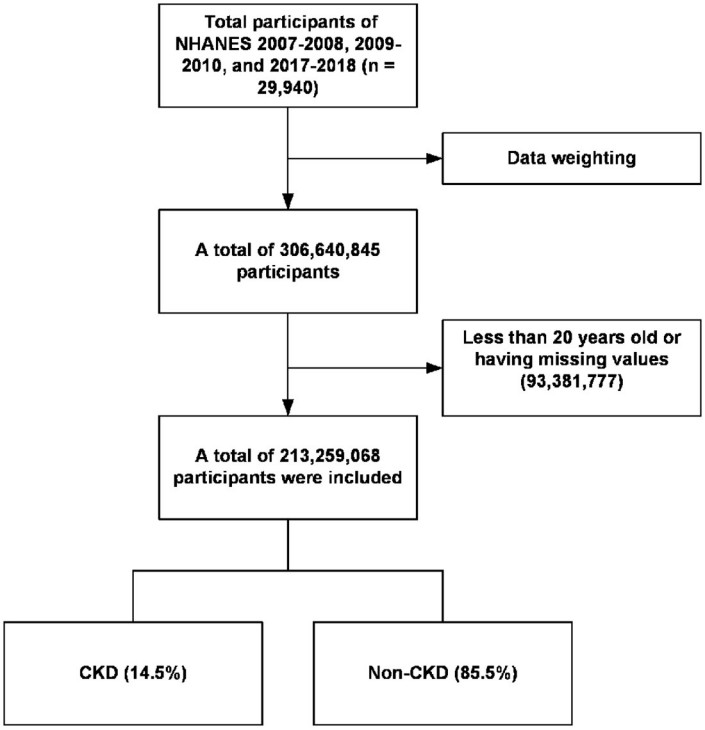

**Fig 1. Flowchart of participants selection.** CKD, chronic kidney disease.

years and above, with 30,889,883 (14.5%) diagnosed with CKD, as shown in Fig 1. The two groups exhibited statistically significant differences in age, sex, ethnicity, height, weight, BMI, SBP, DBP, energy intake, physical smoking status, and prevalence of chronic conditions. Additionally, the intake levels of isoflavones and flavones, eGFR, and uACR differed. Compared with the non-CKD group, the CKD group exhibited older age; higher proportion of women; greater height, weight and BMI; higher SBP; lower DBP; higher prevalence of hyperlipidemia, hypertension, diabetes, and hyperuricemia; and lower daily intake of isoflavones and flavones. Table 1 presents the other results.

## Association between dietary flavonoid intake and CKD

After adjusting for variables including age, sex, body mass index, ethnicity, smoke status, energy intake, physical activity status, hyperuricemia, hyperlipidemia, hypertension, and diabetes mellitus (Table 2), the odds ratios (ORs) for CKD risk in the Q2, Q3, and Q4 groups of total flavonoid intake, compared with that for CKD incidence in the Q1 group, were 0.780 (95% CI: 0.600, 1.015), 0.741 (95% CI: 0.573, 0.957), and 0.716 (95% CI: 0.554, 0.925), respectively. The P value for the trend test was 0.040.

According to the RCS analysis shown in Fig 2, total flavonoid intake exhibited a non-linear relationship with CKD risk (P < 0.001). After multivariate calibration (Table 3), the ORs for decreased eGFR risk in the Q2, Q3, and Q4 groups of total flavonoid intake, compared with that for decreased eGFR incidence in the Q1 group, were 0.529 (95% CI: 0.341, 0.821), 0.354 (95% CI: 0.237, 0.529), and 0.533 (95% CI: 0.329, 0.863), respectively. The P value for the trend test was 0.046. In contrast, there is no significant association between flavonoid intake and the risk of elevated uACR.

**Table 1. Characteristics of study in US population.**

| Variables | Total population (100%) | non-CKD group (85.5%) | CKD group (14.5%) | P value |
|---|---|---|---|---|
| **Age (years)** | 47.49(46.86, 48.12) | 45.23(44.61, 45.82) | 60.9 (59.88, 61.95) | < **0.001** |
| **Age group, *n* (%)** | | | | < **0.001** |
| 20~64 | 73.92(69.34,78.50) | 79.70(78.25,81.15) | 39.79(36.88,42.70) | |
| 65~ | 26.08(24.00,28.17) | 20.30(18.85,21.75) | 60.21(57.30,63.12) | |
| **Sex, *n* (%)** | | | | **0.002** |
| Female | 52.69(49.42,55.96) | 51.96(50.78,53.15) | 57.00(54.03,59.97) | |
| Male | 47.31(44.35,50.26) | 48.04(46.85,49.22) | 43.00(40.03,45.97) | |
| **Ethnicity, *n* (%)** | | | | < **0.001** |
| Black | 10.85(9.22,12.49) | 10.60(8.83,12.37) | 12.34(10.08,14.61) | |
| Mexican | 8.69(6.84,10.54) | 8.88(6.82,10.94) | 7.55(5.82, 9.29) | |
| Others | 12.92(11.19,14.65) | 13.42(11.47,15.38) | 9.97(7.69,12.25) | |
| White | 67.54(60.76,74.32) | 67.10(63.34,70.86) | 70.14(66.16,74.12) | |
| **Education levels, *n* (%)** | | | | 0.486 |
| High school or above | 55.15(50.88,59.42) | 55.17(52.87,57.47) | 56.28(53.19,59.38) | |
| Middle school or below | 44.85(41.39,48.31) | 44.83(42.53,47.13) | 43.72(40.62,46.81) | |
| **Height (cm)** | 168.48(168.19,168.76) | 168.91(168.61,169.21) | 165.90(165.29,166.51) | < **0.001** |
| **Weight (kg)** | 83.13(82.43,83.83) | 82.83(82.11,83.54) | 84.96(83.30,86.62) | **0.017** |
| **BMI (kg/m2)** | 29.19(28.95,29.43) | 28.94(28.71,29.18) | 30.65(30.12,31.19) | < **0.001** |
| **SBP (mm Hg)** | 121.49(120.90,122.07) | 119.85(119.28,120.41) | 131.19(129.59,132.79) | < **0.001** |
| **DBP (mm Hg)** | 70.94(70.27,71.61) | 71.17(70.48,71.87) | 69.54(68.63,70.45) | < **0.001** |
| **PIR, *n* (%)** | | | | 0.834 |
| $\geq 1$ | 86.21(84.58,89.67) | 86.36(85.02,87.69) | 86.18(84.26,88.10) | |
| $< 1$ | 13.79(12.79,13.95) | 13.64(12.31,14.98) | 13.82(11.90,15.74) | |
| **Energy intake (kcal)** | 2136(2108,2165) | 2176(2144,2208) | 1903(1847,1960) | < **0.001** |
| **Physical activity status** | | | | |
| <600 MET min/week | 17.56(15.60,19.16) | 15.50(14.08,16.92) | 23.68(20.49,26.87) | < **0.001** |
| $\geq$600 MET min/week | 82.44(76.13,85.75) | 84.50(83.08,85.92) | 76.32(73.13,79.52) | |
| **Smoking status, n (%)** | | | | < **0.001** |
| Former | 25.01(23.02,26.99) | 23.53(22.06,25.00) | 33.73(31.11,36.35) | |
| Never | 55.85(52.34,59.37) | 56.73(54.53,58.93) | 50.69(47.36,54.02) | |
| Now | 19.14(17.15,21.12) | 19.74(18.31,21.17) | 15.58(13.22,17.95) | |
| **Hyperlipidemia, n (%)** | | | | < **0.001** |
| No | 28.34(26.06,30.62) | 30.18(28.33,32.03) | 17.48(15.17,19.79) | |
| Yes | 71.66(67.08,76.24) | 69.82(67.97,71.67) | 82.52(80.21,84.83) | |
| **Hypertension, n (%)** | | | | < **0.001** |
| No | 62.93(59.39,66.46) | 68.22(66.42,70.02) | 31.69(28.51,34.88) | |
| Yes | 37.07(33.91,40.23) | 31.78(29.98,33.58) | 68.31(65.12,71.49) | |
| **Diabetes, n (%)** | | | | < **0.001** |
| No | 86.16(80.33,91.98) | 89.47(88.68,90.26) | 63.74(60.97,66.50) | |
| Yes | 13.84(12.61,15.08) | 10.53(9.74,11.32) | 36.26(33.50,39.03) | |
| **hyperuricemia, n (%)** | | | | < **0.001** |
| No | 81.73(76.20,87.26) | 84.39(83.29,85.49) | 65.25(62.82,67.68) | |
| Yes | 18.27(16.72,19.82) | 15.61(14.52,16.71) | 34.75(32.32,37.18) | |
| **Isoflavones (mg)** | 2.03(1.73,2.33) | 2.14(1.80,2.48) | 1.37(0.98,1.77) | **0.007** |
| **Anthocyanidins (mg)** | 14.32(12.82,15.82) | 14.50(12.87,16.13) | 13.25(11.14,15.37) | 0.292 |
| **Flavan-3-ols (mg)** | 175.88(161.47,190.28) | 175.98(161.90,190.07) | 175.25(135.21,215.28) | 0.971 |
| **Flavanones (mg)** | 12.44(11.70,13.17) | 12.46(11.73,13.20) | 12.27(10.90,13.64) | 0.765 |

(*Continued*)

**Table 1.** (Continued)

| Variables | Total population (100%) | non-CKD group (85.5%) | CKD group (14.5%) | P value |
|---|---|---|---|---|
| **Flavones (mg)** | 0.96(0.88,1.0 | 0.98(0.89,1.06) | 0.83(0.75,0.92) | **0.015** |
| **Flavonols (mg)** | 18.65(17.98,19.31) | 18.86(18.18,19.54) | 17.39(15.72,19.06) | 0.096 |
| **All flavonoids (mg)** | 224.26(209.27,239.26) | 224.92(210.20,239.65) | 220.37(178.82,261.91) | 0.829 |
| **eGFR (mL/min/1.73m$^2$)** | 94.44(93.46,95.42) | 98.09(97.18,99.00) | 72.25(70.54,73.97) | **< 0.001** |
| **uACR (mg/g)** | 33.72(27.99,39.46) | 7.49(7.29, 7.69) | 191.33(154.68,227.99) | **< 0.001** |

Data are presented as mean for continuous variables or proportions for categorical variables with adjusted 95% confidence interval;

Abbreviations: CKD, chronic kidney disease; PIR, poverty income ratio; BMI, body mass index; SBP, systolic blood pressure; DBP, diastolic blood pressure; eGFR, estimated glomerular filtration rate; MET: metabolic equivalent of task; uACR, urinary albumin-to-creatinine ratio.

## Examining the relationship between dietary flavonoid consumption and CKD through subgroup analyses

Table 4 presents the results of our analysis of the association between dietary flavonoid intake and CKD within stratified groups based on sex, age, race, BMI, educational level, smoking status, hypertension, diabetes, hyperlipidemia, and hyperuricemia. Following multivariate adjustment, the ORs (95% CI) for CKD incidence in Q4 compared with Q1 were 0.571 (0.403, 0.811) for women, 0.674 (0.467, 0.973) for participants more than 65 years old, 0.660 (0.490, 0.888) for Caucasians, 0.531 (0.337, 0.837) for participants with a BMI of <25 kg/m$^2$, 0.731 (0.556, 0.960) for participants with hyperlipidemia, 0.576 (0.399, 0.830) for participants with hypertension, and 0.692 (0.503, 0.953) for participants without diabetes. Among these subgroups, all trend tests yielded *P*-values ≥0.05, except for women, Caucasians, participants with a BMI less than 25 kg/m$^2$, those with hypertension, and those without diabetes. Interaction tests within these subgroups also exhibited *P* > 0.05.

## Relationship between intakes of six flavonoid subclasses and CKD

Following multivariate adjustment, compared with the Q1 group, ORs (95% CI) for CKD risk in the flavonol subgroup were for 0.701 (0.518, 0.947) for Q3 and 0.642 (0.494, 0.835) for Q4, with a *P* value for trend of 0.007. Table 5 presents other results. Fig 3 illustrated the nonlinear trend between the intake of six flavonoid subclasses and the CKD risk.

## Discussion

Based on our analysis of data from NHANES 2007–2010 and 2017–2018, high total flavonoid intake was negatively correlated to CKD risk. Our RCS analysis indicated that a daily flavonoid

**Table 2. Association between dietary flavonoid intake and chronic kidney disease in US population.**

| Total flavonoids (mg per day) | OR (95%CI) | | | P value for trend |
|---|---|---|---|---|
| | Model 1 | Model 2 | Model 3 | |
| Q1 [0, 20.65] | Ref | Ref | Ref | **0.169** |
| Q2 (20.65, 49.74] | 0.798(0.617,1.032) | **0.702(0.548,0.898)** | 0.780(0.600,1.015) | |
| Q3 (49.74, 140.15] | 0.920(0.744,1.137) | **0.699(0.573,0.854)** | **0.741(0.573,0.957)** | |
| Q4 (140.15, 9969.83] | **0.779(0.627,0.966)** | **0.606(0.492,0.747)** | **0.716(0.554,0.925)** | |

Abbreviations: OR, odds ratio; CI, confidence interval; Q1, quartile 1; Q2, quartile 2; Q3, quartile 3; Q4, quartile 4.

Model 1 did not include any covariate. Model 2 was adjusted for age and sex. Model 3 was adjusted for age, sex, body mass index, ethnicity, smoke status, energy intake, physical activity status, hyperuricemia, hyperlipidemia, hypertension, and diabetes mellitus.

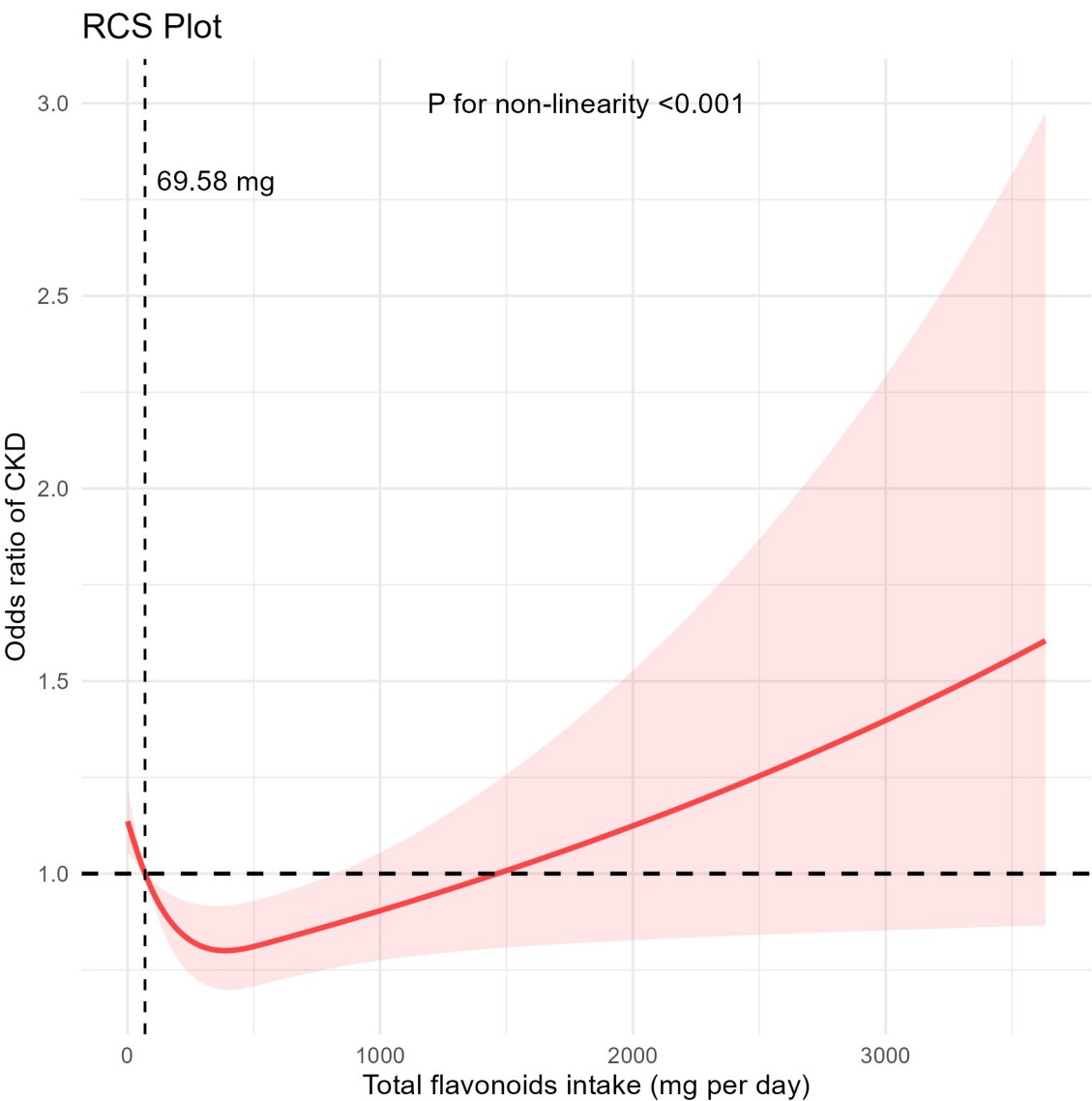

**Fig 2. The non-linear trend between the intake of flavonoids and the risk of CKD using a restricted cubic spline.** Data are presented as odds ratios of CKD (y-axis) and level of flavonoids (mg/d) after adjusted for age, sex, body mass index, ethnicity, smoke status, energy intake, physical activity status, hyperuricemia, hyperlipidemia, hypertension, and diabetes. RCS, restricted cubic spline; CKD, chronic kidney disease.

**Table 3. Association between dietary flavonoid and decreased eGFR/increased uACR in the US population.**

| Outcomes | Total flavonoid (mg per day) | OR (95%CI) | | | | P value for trend |
|---|---|---|---|---|---|---|
| | | Q1 [0, 20.65] | Q2 (20.65, 49.74 | Q3 (49.74, 140.15] | Q4 (140.15, 9969.83] | |
| Decreased eGFR | | ref | **0.529 (0.341,0.821)** | **0.354 (0.237,0.529)** | **0.533 (0.329,0.863)** | **0.046** |
| Elevating uACR | | ref | 0.980 (0.686,1.399) | 1.072 (0.791,1.453) | 0.891 (0.676,1.173) | 0.410 |

Abbreviations: OR, odds ratio; CI, confidence interval; Q1, quartile 1; Q2, quartile 2; Q3, quartile 3; Q4, quartile 4.

Model 1 did not include any covariate. Model 2 was adjusted for age and sex. Model 3 was adjusted for age, sex, body mass index, ethnicity, smoke status, energy intake, physical activity status, hyperuricemia, hyperlipidemia, hypertension, and diabetes mellitus.

**Table 4. Stratified analyses by potential modifiers of the association between dietary flavonoid intake and chronic kidney disease in US population.**

| Subgroup category | Q1 | Q2 | Q3 | Q4 | P value for trend | P value for interaction |
|---|---|---|---|---|---|---|
| | | | OR (95%CI) | | | |
| **Sex** | | | | | | 0.320 |
| Female | ref | 0.721(0.473,1.099) | **0.615(0.428,0.885)** | **0.571(0.403,0.811)** | **0.007** | |
| Male | ref | 0.827(0.529,1.295) | 0.923(0.651,1.310) | 0.957(0.639,1.433) | 0.935 | |
| **Age, years old** | | | | | | 0.856 |
| 20~64 | ref | 0.866(0.588,1.276) | 0.866(0.587,1.279) | 0.803(0.558,1.157) | 0.299 | |
| 65~ | ref | 0.733(0.493,1.090) | 0.615(0.426,0.890) | **0.674(0.467,0.973)** | 0.059 | |
| **Ethnicity** | | | | | | 0.239 |
| Black | ref | 0.714(0.369,1.381) | 0.850(0.482,1.498) | 0.703(0.399,1.238) | 0.213 | |
| White | ref | 0.763(0.555,1.049) | **0.589(0.431,0.807)** | **0.660(0.490,0.888)** | **0.008** | |
| Mexican | ref | 0.933(0.489, 1.780) | 1.480(0.901, 2.432) | 0.681(0.343, 1.353) | 0.261 | |
| Others | ref | 0.882(0.408,1.908) | 1.190(0.582,2.432) | 0.957(0.463,1.978) | 0.902 | |
| **BMI, kg/m$^2$** | | | | | | 0.458 |
| ≥25 | ref | 0.813(0.574,1.152) | 0.734(0.538,1.001) | 0.786(0.579,1.068)) | 0.174 | |
| <25 | ref | 0.644(0.361,1.149) | 0.719(0.428,1.208) | **0.531(0.337,0.837)** | **0.026** | |
| **Smoke status** | | | | | | 0.798 |
| Never | ref | 0.972(0.584,1.620) | 0.831(0.501,1.381) | 0.831(0.501,1.381) | 0.316 | |
| Former | ref | 0.612(0.330,1.135) | 0.576(0.290,1.144) | 0.599(0.311,1.152) | 0.175 | |
| Now | ref | 0.557(0.292,1.062) | 0.793(0.442,1.422) | 0.701(0.428,1.150) | 0.281 | |
| **Hyperlipidemia** | | | | | | 0.932 |
| Yes | ref | 0.758(0.566,1.015) | 0.747(0.557,1.003) | **0.731(0.556,0.960)** | 0.098 | |
| No | ref | 0.780(0.411,1.483) | 0.666(0.321,1.382) | 0.641(0.324,1.269) | 0.199 | |
| **Hypertension** | | | | | | 0.281 |
| Yes | ref | **0.582(0.423,0.800)** | **0.586(0.411,0.836)** | **0.576(0.399,0.830)** | **0.018** | |
| No | ref | 1.121(0.669,1.879) | 0.980(0.662,1.451) | 0.991(0.695,1.413) | 0.748 | |
| **Diabetes mellitus** | | | | | | 0.904 |
| Yes | ref | 0.709(0.445,1.128) | 0.672(0.410,1.102) | 0.803(0.505,1.277) | 0.505 | |
| No | ref | 0.787(0.557,1.113) | 0.767(0.550,1.068) | **0.692(0.503,0.953)** | **0.045** | |
| **Hyperuricemia** | | | | | | 0.890 |
| Yes | ref | 0.644(0.374,1.109) | 0.731(0.403,1.324) | 0.594(0.284,1.245) | 0.228 | |
| No | ref | 0.828(0.579,1.185) | 0.761(0.532,1.090) | 0.763(0.543,1.072) | 0.151 | |

Abbreviations: OR, odds ratio; CI, confidence interval; BMI, body mass index; Q1, quartile 1; Q2, quartile 2; Q3, quartile 3; Q4, quartile 4.

The model was adjusted, if not stratified, for age, sex, body mass index, ethnicity, smoke status, energy intake, physical activity status, hyperuricemia, hyperlipidemia, hypertension, and diabetes. mellitus.

intake of > 69.58 mg is associated with a reduced risk of CKD. However, no significant relationship was observed between higher daily flavonoid intake and reduced CKD risk. Overall, there is a J-shaped relationship between daily flavonoid intake and reduced CKD risk. The P-values from the interaction tests for each subgroup were all greater than 0.05, indicating the robustness of the study findings across all subgroups. In the flavonoid subgroup analysis, we found that the intake of flavanols may play a key role in renal benefits.

Although flavonoids are commonly present in dietary sources and herbal plants, clinical investigations into the relationship between flavonoid and CKD remain relatively scarce. Soy protein containing isoflavones significantly reduces serum creatinine, serum phosphate, urinary protein, and C-reactive protein levels in non-dialysis CKD patients [39]. A statistically significant trend was noted fin the trend analysis, but daily isoflavones intake was not

**Table 5. Relationship between the intakes of six flavonoid subclasses and chronic kidney disease in US adults.**

| Flavonoid subclasses (mg per day) | Q1 | Q2 | Q3 | Q4 | P value for trend |
|---|---|---|---|---|---|
| | | | Adjusted OR (95%CI) | | |
| Isoflavones (mg) | [0, 0] | (0, 0.01] | (0.01, 0.07] | (0.07, 496.00] | |
| | ref | 0.969(0.748,1.256) | 0.893(0.643,1.241) | 0.895(0.689,1.161) | 0.345 |
| Anthocyanidins (mg) | [0, 0.13] | (0.13, 1.91] | (1.91, 9.34] | (9.34, 756.10] | |
| | ref | 1.159(0.890,1.510) | 1.068(0.810,1.408) | 1.058(0.843,1.329) | 0.864 |
| Flavan-3-ols (mg) | [0, 4.61] | (4.61, 13.25] | (13.25, 59.53] | (59.53, 9616.49] | |
| | ref | 1.013(0.716,1.432) | 0.769(0.538,1.098) | 0.811(0.592,1.112) | 0.102 |
| Flavanones (mg) | [0, 0.02] | (0.02, 0.66] | (0.66, 17.73] | (17.173 770.83] | |
| | ref | 0.978(0.776,1.231) | 0.908(0.644,1.280) | 0.962(0.681,1.358) | 0.755 |
| Flavones (mg) | [0, 0.11] | (0.11, 0.35] | (0.35, 0.85] | (0.85, 87.25] | |
| | ref | 0.969(0.748,1.257) | 0.923(0.696,1.222) | 0.867(0.669,1.123) | 0.230 |
| Flavonols (mg) | [0, 4.74] | (4.74, 9.25] | (9.25, 17.33] | (17.33, 348.73] | |
| | ref | 0.711(0.503,1.006) | **0.701(0.518,0.947)** | **0.642(0.494,0.835)** | **0.007** |

Abbreviations: OR, odds ratio; CI, confidence interval; BMI, body mass index; Q1, quartile 1; Q2, quartile 2; Q3, quartile 3; Q4, quartile 4.

All models were adjusted for age, sex, body mass index, ethnicity, smoke status, energy intake, physical activity status, hyperuricemia, hyperlipidemia, hypertension, and diabetes.

significantly associated with CKD incidence. This is a cross-sectional study and does not establish a causal relationship between isoflavones and CKD. Furthermore, isoflavones in the present study were derived from soy and other sources. In a randomized controlled trial (RCT), Rassaf *et al.* demonstrated that daily consumption of cocoa flavanols can improve the vascular endothelial function in patients with end-stage renal disease [21]. Conversely, a CKD population using sodium nitrite and quercetin (a type of flavanol) in Chen and colleagues' research did not yield vascular endothelial benefits [40]. These two studies differed in terms of their study populations, flavanol sources, flavanol dosages, and methods used for evaluating the vascular endothelial function. The present study did not investigate whether flavanol intake could confer vascular benefits to CKD patients. Subgroup analysis in this study revealed that high flavanols intake was significantly and inversely correlated with CKD incidence, with a trend test *P* value of 0.05. This association may be attributed to the vascular endothelial protective, anti-inflammatory, and antioxidant functions inherent in flavonol [41–45]. However, flavanols consumed by the study population primarily originated from everyday foods and beverages, daily flavanol intake was relatively modest. Therefore, whether additional intake of higher doses of flavanols can effectively reduce CKD risk remains unclear. Using data from the NHANES database, Liu *et al.* discovered that daily high flavonoid intake was negatively correlated with the risk of diabetic nephropathy [20]. However, we observed no significant relationship between flavonoid intake and CKD within the subgroup of diabetic patients. Differences in the study populations and endpoint events were observed between the two studies. Liu *et al.'s* study did not diagnose diabetes on the basis of fasting blood glucose and postprandial blood glucose levels, but they used only uACR for the diagnosis of diabetic nephropathy, which may have introduced biases into the study results. Notably, the consumption of flavonoid-rich substances may cause renal function impairment, which is characterized by acute interstitial nephritis and acute tubular necrosis in clinical case reports [46, 47]. Overall, the potential renal benefits of flavonoid consumption in CKD populations remain debatable.

Many basic studies have explored the relationship between flavonoid compounds such as quercetin, silymarin, epimedium, chrysin, and anthocyanins and CKD. Quercetin, a flavonol

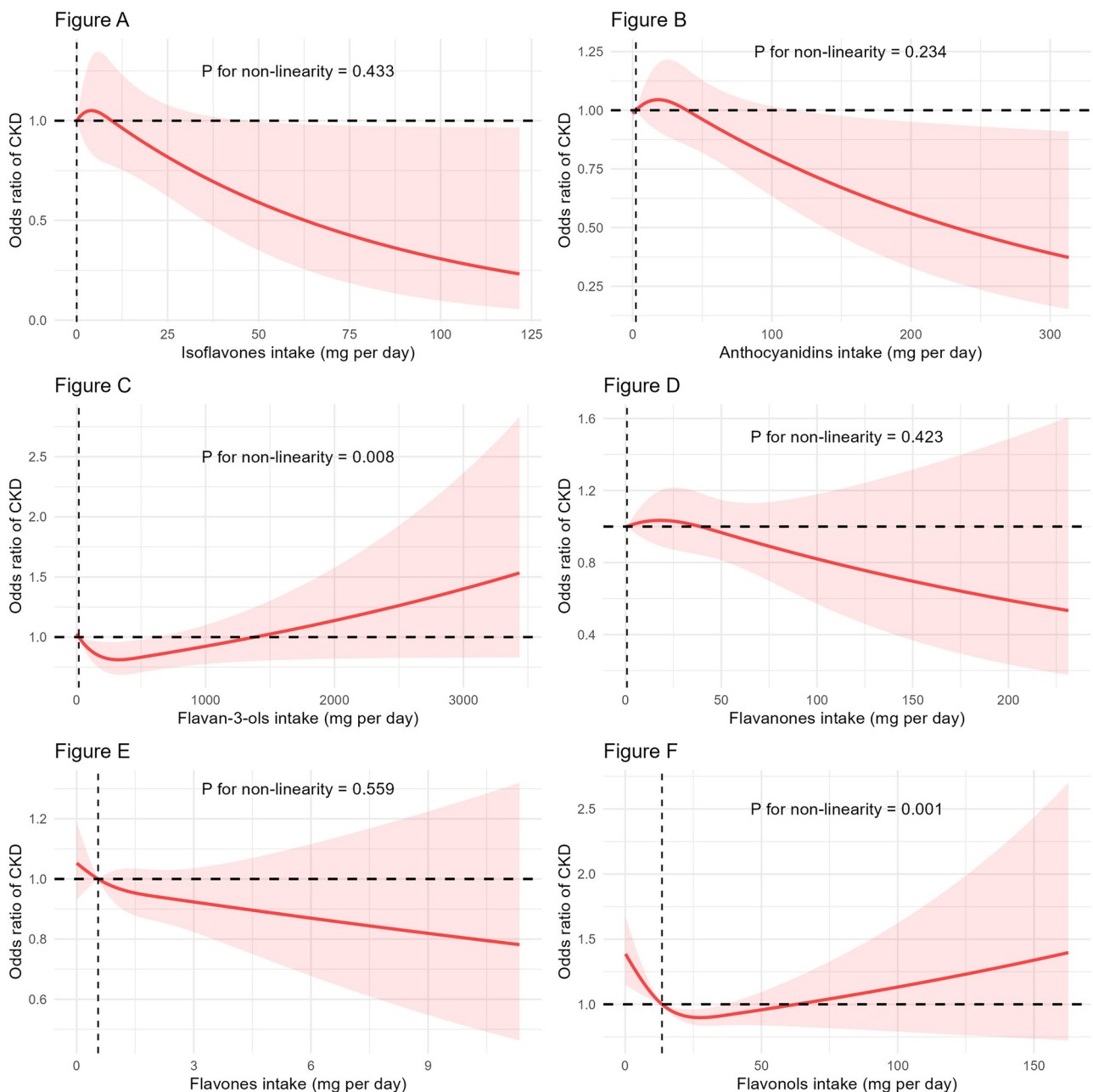

**Fig 3. Utilizing restricted cubic splines to analyze the nonlinear trends between the intake of six flavonoid subclasses and the CKD risk.** CKD, chronic kidney disease.

type, inhibits apoptosis in renal glomerular endothelial cells through the endoplasmic reticulum stress pathway [22]. Furthermore, quercetin possesses anti-inflammatory and antifibrotic effects. It reduces uremic toxin production through the gastrointestinal tract [45, 48, 49]. Flavonoid compounds such as epimedium, chrysin, and anthocyanins may enhance kidney function in CKD patients through anti-inflammatory and antifibrotic mechanisms [23, 25, 50–53]. Of note, flavonoid compound intake might exhibit a dose-dependent relationship with potential benefits for CKD [52, 53]. Similar to quercetin, rutin, another flavanol, demonstrated no

renal benefits [54]. Peng *et al.* found that flavonoid-rich green tea inhibited the clearance of uremia toxins in chronic renal failure mice and elevated the serum creatinine and urea nitrogen concentrations [55]. Silymarin, an isoflavone type, exerted anti-renal fibrotic effects by modulating the transforming growth factor-β signaling pathway in *vitro* and in *vivo* [56, 57]. Additionally, subgroup analysis in this study revealed an association between high flavones intake and a reduced risk of CKD. This may be attributed to potential improvements in renal circulation and alleviation of kidney inflammation [58, 59]. In summary, the flavonoid compounds exert beneficial effects on the kidneys through functions such as antioxidation, antifibrosis, anti-apoptosis, anti-inflammation, and reduction of uremic toxins. However, it is possible that various types and sources of flavonoids may exert different effects on the kidneys in individuals with CKD.

To our best understanding, no study has explored the relationship between CKD and daily flavonoid consumption in a large-scale population. We analyzed data from the NHANES survey to examine the association between flavonoid intake, including its subcategories, and CKD prevalence, in the US population and various subgroups.

This study has certain limitations that warrant acknowledgment. First, the definition of CKD involves abnormal kidney function for a duration of >3 months, constituting a chronic process. However, the average flavonoid intake over 2 days may not accurately represent daily dietary habits, potentially introducing bias into the study findings. Second, the study's participant pool was confined to patients residing in the US, and the relationship between daily flavonoid consumption and CKD among the populations of other countries or regions remains unclear. Third, both flavonoids and their subgroups include various compounds, each with varying biological activities. Consequently, the present study could not specifically elucidate which compound(s) may influence CKD. Fourth, the absence of renal ultrasonography and renal pathological data may have caused an underestimation of CKD incidence, potentially resulting in a bias in the study findings. Fifth, some confounding factors may not have been considered, which could lead to bias. Finally, the excessive daily intake of flavan-3-ols may potentially affect the biological effects of other flavonoid subgroups.

## Conclusions

On analyzing data from NHANES 2007–2008, 2009–2010 and 2017–2018, a potential J-shaped relationship was observed between total flavonoid consumption and CKD risk, with an inflection point at 69.58 mg/d. Our study indicates that a moderate intake of flavonoids may confer renal benefits which may offer novel strategies for CKD treatment.

## Acknowledgments

The authors are grateful to the National Health and Nutrition Examination Survey (NHANES) team for providing the data.

## Author Contributions

**Conceptualization:** Peijia Liu, Wujian Peng.

**Data curation:** Peijia Liu, Wujian Peng.

**Formal analysis:** Peijia Liu, Wujian Peng.

**Funding acquisition:** Wujian Peng, Guixia Li.

**Investigation:** Peijia Liu, Wujian Peng.

**Methodology:** Peijia Liu, Wujian Peng.

**Project administration:** Peijia Liu, Wujian Peng.

**Resources:** Peijia Liu, Wujian Peng.

**Software:** Peijia Liu, Wujian Peng, Guixia Li.

**Supervision:** Feng Hu, Guixia Li.

**Validation:** Feng Hu, Guixia Li.

**Visualization:** Feng Hu, Guixia Li.

**Writing – original draft:** Feng Hu, Guixia Li.

**Writing – review & editing:** Feng Hu, Guixia Li.

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
