## [Decision Letter · Decision Letter 0]

21 May 2024

PONE-D-24-13778Association between dietary intake of flavonoid and chronic kidney disease in US adults: Evidence from NHANES 2007-2008, 2009-2010, and 2017-2018PLOS ONE

Dear Dr. Li,

Thank you for submitting your manuscript to PLOS ONE. After careful consideration, we feel that it has merit but does not fully meet PLOS ONE’s publication criteria as it currently stands. Therefore, we invite you to submit a revised version of the manuscript that addresses the points raised during the review process.

We look forward to receiving your revised manuscript.

Kind regards,

Tauqeer Hussain Mallhi, Ph.D

Academic Editor

PLOS ONE

Journal Requirements:

 [This work was supported by Project funded by Shenzhen Third People's Hospital (No. G2022030) and Shenzhen Science and technology planning project (No.JCYJ20220530163003007].  

Additional Editor Comments:

Dear Authors, Thank you for submitting in Plos One. Your manuscript has been assessed by relevant experts in the field. They found several methodological concerns in the manuscript. The authors are advised to consider the comments of the reviewers specifically adjustment of confounders, data collection procedure and bias.

Reviewers' comments:

Reviewer's Responses to Questions

**Comments to the Author**

1. Is the manuscript technically sound, and do the data support the conclusions?

Reviewer #1: Yes

Reviewer #2: Partly

Reviewer #3: Partly

Reviewer #4: Yes

2. Has the statistical analysis been performed appropriately and rigorously? 

Reviewer #1: I Don't Know

Reviewer #2: I Don't Know

Reviewer #3: I Don't Know

Reviewer #4: Yes

3. Have the authors made all data underlying the findings in their manuscript fully available?

Reviewer #1: Yes

Reviewer #2: Yes

Reviewer #3: Yes

Reviewer #4: Yes

4. Is the manuscript presented in an intelligible fashion and written in standard English?

Reviewer #1: Yes

Reviewer #2: Yes

Reviewer #3: Yes

Reviewer #4: Yes

5. Review Comments to the Author

Reviewer #1: It is a very interesting article. It is well-designed although cross-sectional. However, I think a professional should review the statistical analyses. The figures are vague. It would be nice to analyze the effects according to different stages of CKD, both GFR stages and albuminuria stages.

Reviewer #2: The article aims to find an association between dietary intake of flavonoid and CKD in US adults using the NHANES data between 2007-2008. 2009-2010 and 2017-2018. The methodology is complex and requires more explanation. For instance: in lines 53-55 the authors write "This study employed a cross-sectional design and used data from three cycles of the continuous NHANES: 2007–2008, 2009–2010, and 2017–2018. After weighting, the findings of this study are representative of 306,640,845 US residents". "Weighting'' needs to be explained clearly so the reader is left with a clear understanding of the methodology. This also applies to line 135 of the manuscript "Post-weighting, the data in this study represented 213,259,068 US adults aged 20 years..." . ''Post-weighting'' should also be clearly explained.

The text in lines 163-165: "Among these subgroups, all trend tests yielded P < 0.05, except for participants older than 65 years and those with hyperlipidemia. Interaction tests within these subgroups also exhibited P > 0.05". Table 3 does not align with the text here, adults >65 years had a p-value for trend of 0.059 and a p-value for interaction of 0.856. Those with hyperlipidemia also had a p-value >0.05. Please clarify these seeming discrepancies. Also review lines 167-168 and Table 4 to ensure the text aligns with the table.

Overall, the article is well written, and could evoke further research on the use of flavinoids and its association with CKD.

Reviewer #3: The authors conducted a cross-sectional study using the NHANES database on flavonoid intake and the presence of chronic kidney disease among adults in the United States. Flavonoid intake was negatively associated with risk of chronic kidney disease, but there was no statistically significant association between flavonoid intake and risk of chronic kidney disease.

While this study is informative in that it focuses on a familiar nutrient and analyzes its preventive effect on chronic kidney disease, the methods used to collect and interpret the data are very difficult. I offer my opinion on this paper.

Major

1. The authors do not clearly state the primary outcome in the Methods section. The reader cannot understand what the authors are trying to clarify.

2. The authors' data collection method is very vague, and it is difficult to determine whether a quantitative evaluation can be made with a questionnaire. The authors' method of converting disparate data into continuous variables and then calculating averages from them is not appropriate.

3. Except when consuming nutrient extracts such as supplements, nutrients obtained from food are complex and interact, making it difficult to consider some nutrients obtained from food in isolation. It is difficult to rule out various confounding factors, such as alcohol in wine or caffeine in tea.

4. It is difficult to dispel the bias that people who consume more flavonoids are more health conscious. Therefore, it is difficult to prove causality outside of intervention studies.

Minor

The resolution of Fig. 2 is poor.

Reviewer #4: This is a cross sectional, registry based study on the association between dietary intake of flavonoid and chronic kidney disease. The study population is well defined and representative for the whole US population. Food consumption is based on 24-hour dietary recall questionnaire and CKD is defined as an e-GFR of <60 ml/min/1.73m2 or as UAC of >30 mg/g.

The research question is interesting and important, flavenoids have been associated with better cardiometabolic health and have shown to be renoprotective in animal models. There are few clinical studies that have adressed the potential beneficial effect of dietary flavenoid on kidney function/disease.

The authors find that total flavonoid intake is negatively associated with CKD and that the relationship is non-linear with an inflection point at about 70 mg/d. The authors conclude that moderate intake of flavonoids may confer renal benefits and offer novel strategies for CKD treatment. They also looked at the six flavenoid subgroups and found only significant negative association between flavonol and CKD.

These results are in line with other studies that find beneficial effect of flavenoids on renal function.

I have the following questions or comments:

1. The study population: In Methods, lines 55 and 57, the study is said to represent 306,640,845 US residents after weighting, and after exclusion of individuals under 20 years and those with missing data, the total is 13,259,068. On the other hand, in Results, line 135 the number is 213,259,068 US adults >20 years of age. Please comment on this discrepancy, and also provide the actual number of participants, ie before the weighting.

2. The outcome variable is CKD, based on an e-GFR of <60 ml/min/1.73m2 or as UAC of >30 mg/g. According to these criteria, 14.5% of the study population has CKD. In recent years, studies have shown that this may overestimate the prevalence of CKD, and that age-adapted e-GFR may be more appropriate and would consider the physiologic age-related decline in GFR. Overestimation of eGFR in the elderly could bias the outcome of the study. Please comment on this.

3. I propose that the authors describe the types of food and beverages that belong to the various subclass of flavonoids in a table. This will give the reader a better picture of the results.

4. There is a great variation in the intake of various subtypes of flavonoids (table 4). Flavonols are abundant and the only subtype that has a significantly negative association with CKD. Flavan-3-ols are also abundant in the food but do not confer renal protection. Comment on that please. In Discussion, lines 253-254, the authors say: “the excessive daily intake of flavan-3-ols may potentially affect the biological effects of other flavonoid subgroups”. How? Has this to do with the J shaped curve?

5. In Discussion, line 218 and more, Quercetin is discussed and said to be flavanol, but should be flavonol.

6. PLOS authors have the option to publish the peer review history of their article (what does this mean?). If published, this will include your full peer review and any attached files.

Reviewer #1: No

Reviewer #2: **Yes: **Ngozi Virginia Aikpokpo

Reviewer #3: No

Reviewer #4: No

---

## [Author Response · Author response to Decision Letter 0]

14 Jun 2024

We have responded to the comments from the editors and reviewers on a point-by-point basis. For detailed information, please refer to the document "Response to Reviewers".

---

## [Decision Letter · Decision Letter 1]

30 Jul 2024

Association between dietary intake of flavonoid and chronic kidney disease in US adults: Evidence from NHANES 2007-2008, 2009-2010, and 2017-2018

PONE-D-24-13778R1

Dear Dr. Li,

We’re pleased to inform you that your manuscript has been judged scientifically suitable for publication and will be formally accepted for publication once it meets all outstanding technical requirements.

Kind regards,

Tauqeer Hussain Mallhi, Ph.D

Academic Editor

PLOS ONE

Additional Editor Comments (optional):

Dear Authors, thank you for revising the manuscript.

Reviewers' comments:

Reviewer's Responses to Questions

**Comments to the Author**

1. If the authors have adequately addressed your comments raised in a previous round of review and you feel that this manuscript is now acceptable for publication, you may indicate that here to bypass the “Comments to the Author” section, enter your conflict of interest statement in the “Confidential to Editor” section, and submit your "Accept" recommendation.

Reviewer #1: All comments have been addressed

Reviewer #2: All comments have been addressed

Reviewer #3: All comments have been addressed

2. Is the manuscript technically sound, and do the data support the conclusions?

Reviewer #1: Yes

Reviewer #2: Yes

Reviewer #3: Yes

3. Has the statistical analysis been performed appropriately and rigorously? 

Reviewer #1: I Don't Know

Reviewer #2: I Don't Know

Reviewer #3: Yes

4. Have the authors made all data underlying the findings in their manuscript fully available?

Reviewer #1: Yes

Reviewer #2: Yes

Reviewer #3: Yes

5. Is the manuscript presented in an intelligible fashion and written in standard English?

Reviewer #1: Yes

Reviewer #2: Yes

Reviewer #3: Yes

6. Review Comments to the Author

Reviewer #1: (No Response)

Reviewer #2: The authors have addressed previous concerns. The methodology is still complex but the authors have attempted to explain this. They have also noted possible confounders to the study.

The study can form a basis for exploring the use of flavonoid containing foods as a possible means of preventing CKD. Further studies would have to quantify the amount used so as to adequately explain "moderate or high" intake.

Reviewer #3: The authors have responded to my peer review comments in all sincerity.　The authors have no objection to their paper being accepted.

7. PLOS authors have the option to publish the peer review history of their article (what does this mean?). If published, this will include your full peer review and any attached files.

Reviewer #1: No

Reviewer #2: **Yes: **Ngozi Virginia Aikpokpo

Reviewer #3: No

---

## [Editor Report · Acceptance letter]

15 Aug 2024

PONE-D-24-13778R1 

PLOS ONE

Dear Dr. Li, 

I'm pleased to inform you that your manuscript has been deemed suitable for publication in PLOS ONE. Congratulations! Your manuscript is now being handed over to our production team.

Kind regards, 

on behalf of

Dr. Tauqeer Hussain Mallhi 

Academic Editor

PLOS ONE